# A Convolutional Approach to Learning Time Series Similarity

## Abstract

Computing distances between examples is at the core of many learning algorithms for time series. Consequently, a great deal of work has gone into designing effective time series distance measures. We present Jiffy, a simple and scalable distance metric for multivariate time series. Our approach is to reframe the task as a representation learning problem—rather than design an elaborate distance function, we use a CNN to learn an embedding such that the Euclidean distance is effective. By aggressively max-pooling and downsampling, we are able to construct this embedding using a highly compact neural network. Experiments on a diverse set of multivariate time series datasets show that our approach consistently outperforms existing methods.

## 1 Introduction

Measuring distances between examples is a fundamental component of many classification, clustering, segmentation and anomaly detection algorithms for time series (Rakthanmanon et al., 2012; Schäfer, 2014; Begum et al., 2015; Dau et al., 2016). Because the distance measure used can have a significant effect on the quality of the results, there has been a great deal of work developing effective time series distance measures (Ganeshapillai & Guttag, 2011; Keogh et al., 2005; Bagnall et al., 2016; Begum et al., 2015; Ding et al., 2008). Historically, most of these measures have been hand-crafted. However, recent work has shown that a learning approach can often perform better than traditional techniques (Do et al., 2017; Mei et al., 2016; Che et al., 2017).

We introduce a metric learning model for multivariate time series. Specifically, by learning to embed time series in Euclidean space, we obtain a metric that is both highly effective and simple to implement using modern machine learning libraries. Unlike many other deep metric learning approaches for time series, we use a convolutional, rather than a recurrent, neural network, to construct the embedding. This choice, in combination with aggressive maxpooling and downsampling, results in a compact, accurate network.

Using a convolutional neural network for metric learning *per se* is not a novel idea (Oh Song et al., 2016; Schroff et al., 2015); however, time series present a set of challenges not seen together in other domains, and how best to embed them is far from obvious. In particular, time series suffer from:

1. *A lack of labeled data*. Unlike text or images, time series cannot typically be annotated post-hoc by humans. This has given rise to efforts at unsupervised labeling (Blalock & Guttag, 2016), and is evidenced by the small size of most labeled time series datasets. Of the 85 datasets in the UCR archive (Chen et al., 2015), for example, the largest dataset has fewer than 17000 examples, and many have only a few hundred.
2. *A lack of large corpora*. In addition to the difficulty of obtaining labels, most researchers have no means of gathering even *unlabeled* time series at the same scale as images, videos, or text. Even the largest time series corpora, such as those on Physiobank (Goldberger et al., 2000), are tiny compared to the virtually limitless text, image, and video data available on the web.
3. *Extraneous data*. There is no guarantee that the beginning and end of a time series correspond to the beginning and end of any meaningful phenomenon. I.e., examples of the class or pattern of interest may take place in only a small interval within a much longer time series. The rest of the time series may be noise or transient phenomena between meaningful events (Rakthanmanon et al., 2011; Hao et al., 2013).

4. *Need for high speed*. One consequence of the presence of extraneous data is that many time series algorithms compute distances using every window of data within a time series (Mueen et al., 2009; Blalock & Guttag, 2016; Rakthanmanon et al., 2011). A time series of length $T$ has $O(T)$ windows of a given length, so it is essential that the operations done at each window be efficient.

As a result of these challenges, an effective time series distance metric must exhibit the following properties:

- Efficiency: Distance measurement must be fast, in terms of both training time and inference time.

- Simplicity: As evidenced by the continued dominance of the Dynamic Time Warping (DTW) distance (Sakoe & Chiba, 1978) in the presence of more accurate but more complicated rivals, a distance measure must be simple to understand and implement.

- Accuracy: Given a labeled dataset, the metric should yield a smaller distance between similarly labeled time series. This behavior should hold even for small training sets.

Our primary contribution is a time series metric learning method, *Jiffy*, that exhibits all of these properties: it is fast at both training and inference time, simple to understand and implement, and consistently outperforms existing methods across a variety of datasets.

We introduce the problem statement and the requisite definitions in Section 2. We summarize existing state-of-the-art approaches (both neural and non-neural) in Section 3 and go on to detail our own approach in Section 4. We then present our results in Section 5. The paper concludes with implications of our work and avenues for further research.

## 2 PROBLEM DEFINITION

We first define relevant terms, frame the problem, and state our assumptions.

**Definition 2.1.** *Time Series* A $D$-variable time series $X$ of length $T$ is a sequence of real-valued vectors $\mathbf{x}_1, \ldots, \mathbf{x}_T, \mathbf{x}_i \in \mathbb{R}^D$. If $D = 1$, we call $X$ "univariate", and if $D > 1$, we call $X$ "multivariate." We denote the space of possible $D$-variable time series $\mathcal{T}^D$.

**Definition 2.2.** *Distance Metric* A distance metric is defined a distance function $d : \mathcal{S} \times \mathcal{S} \to \mathbb{R}$ over a set of objects $\mathcal{S}$ such that, for any $x, y \in \mathcal{S}$, the following properties hold:

- *Symmetry:* $d(x, y) = d(y, x)$

- *Non-negativity:* $d(x, y) \geq 0$

- *Triangle Inequality:* $d(x, z) + d(y, z) \geq d(x, z)$

- *Identity of Indiscernibles:* $x = y \Leftrightarrow d(x, y) = 0$

Our approach to learning a metric is to first learn an embedding into a fixed-size vector space, and then use the Euclidean distance on the embedded vectors to measure similarity. Formally, we learn a function $f : \mathcal{T}^D \to \mathbb{R}^N$ and compute the distance between time series $X, Y \in \mathcal{T}^D$ as:

$$d(X, Y) \triangleq \|f(X) - f(Y)\|_2 \tag{1}$$

### 2.1 ASSUMPTIONS

Jiffy depends on two assumptions about the time series being embedded. First, we assume that all time series are primarily "explained" by one class. This means that we do not consider multi-label tasks or tasks wherein only a small subsequence within each time series is associated with a particular label, while the rest is noise or phenomena for which we have no class label. This assumption is implicitly made by most existing work (Hu et al., 2013) and is satisfied whenever one has recordings of individual phenomena, such as gestures, heartbeats, or actions.

The second assumption is that the time series dataset is not too small, in terms of either number of time series or their lengths. Specifically, we do not consider datasets in which the longest time series is of length $T < 40$ or the number of examples per class is less than 25. The former number is the

smallest number such that our embedding will not be longer than the input in the univariate case, while the latter is the smallest number found in any of our experimental datasets (and therefore the smallest on which we can claim reasonable performance).

For datasets too small to satisfy these constraints, we recommend using a traditional distance measure, such as Dynamic Time Warping, that does not rely on a learning phase.

## 3 RELATED WORK

### 3.1 HAND-CRAFTED DISTANCE MEASURES

Historically, most work on distance measures between time series has consisted of hand-crafted algorithms designed to reflect prior knowledge about the nature of time series. By far the most prevalent is the Dynamic Time Warping (DTW) distance (Sakoe & Chiba, 1978). This is obtained by first aligning two time series using dynamic programming, and then computing the Euclidean distance between them. DTW requires time quadratic in the time series' length in the worst case, but is effectively linear time when used for similarity search; this is thanks to numerous lower bounds that allow early abandoning of the computation in almost all cases (Rakthanmanon et al., 2012).

Other handcrafted measures include the Uniform Scaling Distance (Keogh, 2003), the Scaled Warped Matching Distance (Fu et al., 2008), the Complexity-Invariant Distance (Batista et al., 2011), the Shotgun Distance (Schäfer, 2014), and many variants of DTW, such as weighted DTW (Ganeshapillai & Guttag, 2011), DTW-A (Shokoohi-Yekta et al., 2015), and global alignment kernels (Cuturi, 2011). However, nearly all of these measures are defined only for univariate time series, and generalizing them to multivariate time series is not trivial (Shokoohi-Yekta et al., 2015).

### 3.2 HAND-CRAFTED REPRESENTATIONS

In addition to hand-crafted functions of raw time series, there are numerous hand-crafted representations of time series. Perhaps the most common are Symbolic Aggregate Approximation (SAX) (Lin et al., 2003) and its derivatives (Camerra et al., 2010; Senin & Malinchik, 2013). These are discretization techniques that low-pass filter, downsample, and quantize the time series so that they can be treated as strings. Slightly less lossy are Adaptive Piecewise Constant Approximation (Keogh et al., 2001a), Piecewise Aggregate Approximation (Keogh et al., 2001b), and related methods, which approximate time series as sequences of low-order polynomials.

The most effective of these representations tend to be extremely complicated; the current state-of-the-art (Schäfer & Leser, 2017), for example, entails windowing, Fourier transformation, quantization, bigram extraction, and ANOVA F-tests, among other steps. Moreover, it is not obvious how to generalize them to multivariate time series.

### 3.3 METRIC LEARNING FOR TIME SERIES

A promising alternative to hand-crafted representations and distance functions for time series is metric learning. This can take the form of either learning a distance function directly or learning a representation that can be used with an existing distance function.

Among the most well-known methods in the former category is that of (Ratanamahatana & Keogh, 2004a), which uses an iterative search to learn data-dependent constraints on DTW alignments. More recently, Mei et al. (2016) use a learned Mahalanobis distance to improve the accuracy of DTW. Both of these approaches yield only a pseudometric, which does not obey the triangle inequality. To come closer to a true metric, Che et al. (2017) combined a large-margin classification objective with a sampling step (even at test time) to create a DTW-like distance that obeys the triangle inequality with high probability as the sample size increases.

In the second category are various works that learn to embed time series into Euclidean space. Pei et al. (2016) use recurrent neural networks in a Siamese architecture (Bromley et al., 1994) to learn an embedding; they optimize the embeddings to have positive inner products for time series of the same class but negative inner products for those of different classes. A similar approach that does not require class labels is that of Arnaud et al. (2017). This method trains a Siamese, single-layer

CNN to embed time series in a space such that the pairwise Euclidean distances approximate the pairwise DTW distances. Lei et al. (2017) optimize a similar objective, but do so by sampling the pairwise distances and using matrix factorization to directly construct feature representations for the training set (i.e., with no model that could be applied to a separate test set).

These methods seek to solve much the same problem as Jiffy but, as we show experimentally, produce metrics of much lower quality.

## 4 METHOD

We learn a metric by learning to embed time series into a vector space and comparing the resulting vectors with the Euclidean distance. Our embedding function is takes the form of a convolutional neural network, shown in Figure 1. The architecture rests on three basic layers: a convolutional layer, maxpooling layer, and a fully connected layer.

The convolutional layer is included to learn the appropriate subsequences from the input. The network employs one-dimensional filters convolved over all time steps, in contrast to traditional two-dimensional filters used with images. We opt for one-dimensional filters because time series data is characterized by infrequent sampling. Convolving over each of the variables at a given timestep has little intuitive meaning in developing an embedding when each step measurement has no coherent connection to time. For discussion regarding the mathematical connection between a learned convolutional filter and traditional subsequence-based analysis of time series, we direct the reader to (Cui et al., 2016).

The maxpooling layer allows the network to be resilient to translational noise in the input time series. Unlike most existing neural network architectures, the windows over which we max pool are defined as percentages of the input length, not as constants. This level of pooling allows us to heavily downsample and denoise the input signal and is fed into the final fully connected layer.

We downsample heavily after the filters are applied such that each time series is reduced to a fixed size. We do so primarily for efficiency—further discussion on parameter choice for Jiffy may be found in Section 6.

We then train the network by appending a softmax layer and using cross-entropy loss with the ADAM (Kingma & Ba, 2014) optimizer. We experimented with more traditional metric learning loss functions, rather than a classification objective, but found that they made little or no difference while adding to the complexity of the training procedure; specific loss functions tested include several variations of Siamese networks (Bromley et al., 1994; Pei et al., 2016) and the triplet loss (Hoffer & Ailon, 2015).

### 4.1 COMPLEXITY ANALYSIS

For ease of comparison to more traditional distance measures, such as DTW, we present an analysis of Jiffy's complexity.

Let $T$ be the length of the $D$-variable time series being embedded, let $F$ be the number of length $K$ filters used in the convolutional layer, and Let $L$ be the size of the final embedding. The time to apply the convolution and ReLU operations is $\Theta(TDFK)$. Following the convolutional layer, the maxpooling and downsampling require (T2DF) time if implemented naively, but (TDF) if an intelligent sliding max function is used, such as that of (Lemire, 2006). Finally, the fully connected layer, which constitutes the embedding, requires $\Theta(TDFL)$ time.

The total time to generate the embedding is therefore $\Theta(TDF(K + L))$. Given the embeddings, computing the distance between two time series requires $\Theta(L)$ time. Note that $T$ no longer appears in either expression thanks to the max pooling.

With $F = 16$, $K = 5$, $L = 40$, this computation is dominated by the fully connected layer. Consequently, when $L \ll T$ and embeddings can be generated ahead of time, this enables a significant speedup compared to operating on the original data. Such a situation would arise, e.g., when performing a similarity search between a new query and a fixed or slow-changing database (Blalock

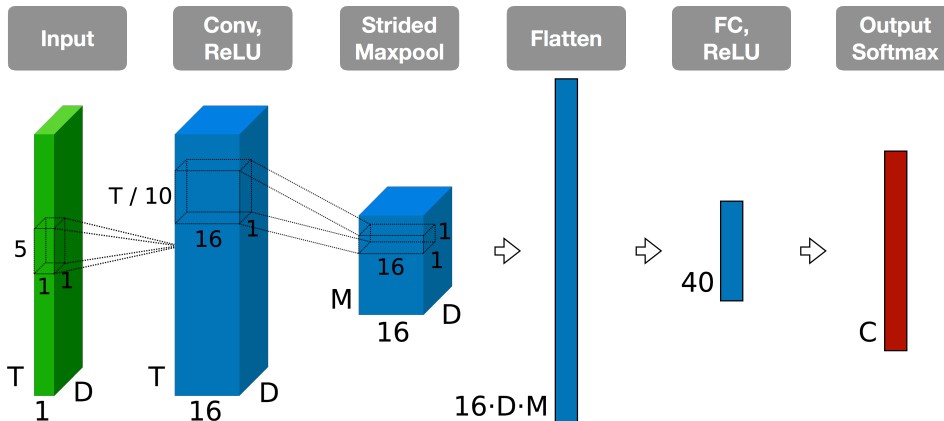

**Figure 1: Architecture of the proposed model. A single convolutional layer extracts local features from the input, which a strided maxpool layer reduces to a fixed-size vector. A fully connected layer with ReLU activation carries out further, nonlinear dimensionality reduction to yield the embedding. A softmax layer is added at training time.**

& Guttag, 2017). When both embeddings must be computed on-the-fly, our method is likely to be slower than DTW and other traditional approaches.

## 5 EXPERIMENTS

Before describing our experiments, we first note that, to ensure easy reproduction and extension of our work, all of our code is freely available.[1] All of the datasets used are public, and we provide code to clean and operate on them.

We evaluate Jiffy-produced embeddings through the task of 1-nearest-neighbor classification, which assesses the extent to which time series sharing the same label tend to be nearby in the embedded space. We choose this task because it is the most widely used benchmark for time series distance and similarity measures (Ding et al., 2008; Bagnall et al., 2016).

### 5.1 DATASETS

To enable direct comparison to existing methods, we benchmark Jiffy using datasets employed by Mei et al. (2016). These datasets are taken from various domains and exhibit high variability in the numbers of classes, examples, and variables. We briefly describe each dataset below, and summarize statistics about each in Table 1.

**Table 1: Summary of Multivariate Time Series Datasets.**

| Dataset | # Variables | # Classes | Length | # Time Series |
|---|---|---|---|---|
| Libras | 2 | 15 | 45 | 360 |
| AUSLAN | 22 | 25 | 47-95 | 675 |
| CharacterTrajectories | 3 | 20 | 109-205 | 2858 |
| ArabicDigits | 13 | 10 | 4 - 93 | 8800 |
| ECG | 2 | 2 | 39 - 152 | 200 |
| Wafer | 6 | 2 | 104 - 198 | 1194 |

- **ECG**: Electrical recordings of normal and abnormal heartbeats, as measured by two electrodes on the patients' chests.

- **Wafer**: Sensor data collected during the manufacture of semiconductor microelectronics, where the time series are labeled as normal or abnormal.

---

[1]http://smarturl.it/jiffy

- **AUSLAN**: Hand and finger positions during the performance of various signs in Australian Sign Language, measured via instrumented gloves.
- **Trajectories**: Recordings of pen (x,y) position and force application as different English characters are written with a pen.
- **Libras**: Hand and arm positions during the performance of various signs in Brazilian Sign Language, extracted from videos.
- **ArabicDigits**: Audio signals produced by utterances of Arabic digits, represented by Mel-Frequency Cepstral Coefficients.

## 5.2 COMPARISON APPROACHES

We compare to recent approaches to time series metric learning, as well as popular means of generalizing DTW to the multivariate case:

1. **MDDTW** (Mei et al., 2016) - MDDTW compares time series using a combination of DTW and the Mahalanobis distance. It learns the precision matrix for the latter using a triplet loss.
2. **Siamese RNN** (Pei et al., 2016) - The Siamese RNN feeds each time series through a recurrent neural network and uses the hidden unit activations as the embedding. It trains by feeding pairs of time series through two copies of the network and computing errors based on their inner products in the embedded space.
3. **Siamese CNN** The Siamese CNN is similar to the Siamese RNN, but uses convolutional, rather than recurrent, neural networks. This approach has proven successful across several computer vision tasks (Bromley et al., 1994; Taigman et al., 2014).
4. **DTW-I**, **DTW-D** - As pointed out by Shokoohi-Yekta et al. (2015), there are two straightforward ways to generalize DTW to multivariate time series. The first is to treat the time series as $D$ independent sequences of scalars (DTW-I). In this case, one computes the DTW distance for each sequence separately, then sums the results. The second option is to treat the time series as one sequence of vectors (DTW-D). In this case, one runs DTW a single time, with elementwise distances equal to the squared Euclidean distances between the $D$-dimensional elements.
5. **Zero Padding** - One means of obtaining a fixed-size vector representation of a multivariate time series is to zero-pad such that all time series are the same length, and then treat the "flattened" representation as a vector.
6. **Upsampling** - Like Zero Padding, but upsamples to the length of the longest time series rather than appending zeros. This approach is known to be effective for univariate time series (Ratanama-hatana & Keogh, 2004b).

## 5.3 ACCURACY

As shown in Table 2, we match or exceed the performance of all comparison methods on each of the six datasets. Although it is not possible to claim statistical significance in the absence of more datasets (see Demsar (2006)), the average rank of our method compared to others is higher than its closest competitors at 1.16. The closest second, DTW-I, has an average rank of 3.33 over these six datasets.

Not only does Jiffy attain higher classification accuracies than competing methods, but the method also remains consistent in its performance across datasets. This can most easily be seen through the standard deviation in classification accuracies across datasets for each method. Jiffy's standard deviation in accuracy (0.026) is approximately a third of DTWI's (0.071). The closest method in terms of variance is MDDTW with a standard deviation of 0.042 , which exhibits a much lower rank than our method. This consistency suggests that Jiffy generalizes well across domains, and would likely remain effective on other datasets not tested here.

## 6 HYPERPARAMETER EFFECTS

A natural question when considering the performance of a neural network is whether, or to what extent, the hyperparameters must be modified to achieve good performance on a new dataset. In

**Table 2: 1NN Classification Accuracy. The proposed method equals or exceeds the accuracies of all others on every dataset.**

| Dataset | Jiffy | MDDTW | DTW-D | DTW-I | Siamese CNN | Siamese RNN | Zero Pad | Upsample |
|---|---|---|---|---|---|---|---|---|
| ArabicDigits | **0.974** | 0.969 | 0.963 | **0.974** | 0.851 | 0.375 | 0.967 | 0.898 |
| AUSLAN | **1.000** | 0.959 | 0.900 | **1.000** | **1.000** | **1.000** | **1.000** | **1.000** |
| ECG | **0.925** | 0.865 | 0.825 | 0.810 | 0.756 | 0.659 | 0.820 | 0.820 |
| Libras | **1.000** | 0.908 | 0.905 | 0.979 | 0.280 | 0.320 | 0.534 | 0.534 |
| Trajectories | **0.979** | 0.961 | 0.956 | 0.972 | 0.933 | 0.816 | 0.936 | 0.948 |
| Wafer | **0.992** | 0.988 | 0.984 | 0.861 | 0.968 | 0.954 | 0.945 | 0.936 |
| Mean Rank | **1.67** | 3.67 | 4.67 | 3.33 | 6.0 | 6.5 | 4.17 | 4.5 |

this section, we explore the robustness of our approach with respect to the values of the two key parameters: embedding size and pooling percentage. We do this by learning metrics for a variety of parameter values for ten data sets from the UCR Time Series Archive (Chen et al., 2015), and evaluating how classification accuracy varies.

## 6.1 EMBEDDING SIZE

Figure 2.*left* shows that even a few dozen neurons are sufficient to achieve peak accuracy. As a result, an embedding layer of 40 neurons is sufficient and leads to an architecture that is compact enough to run on a personal laptop.

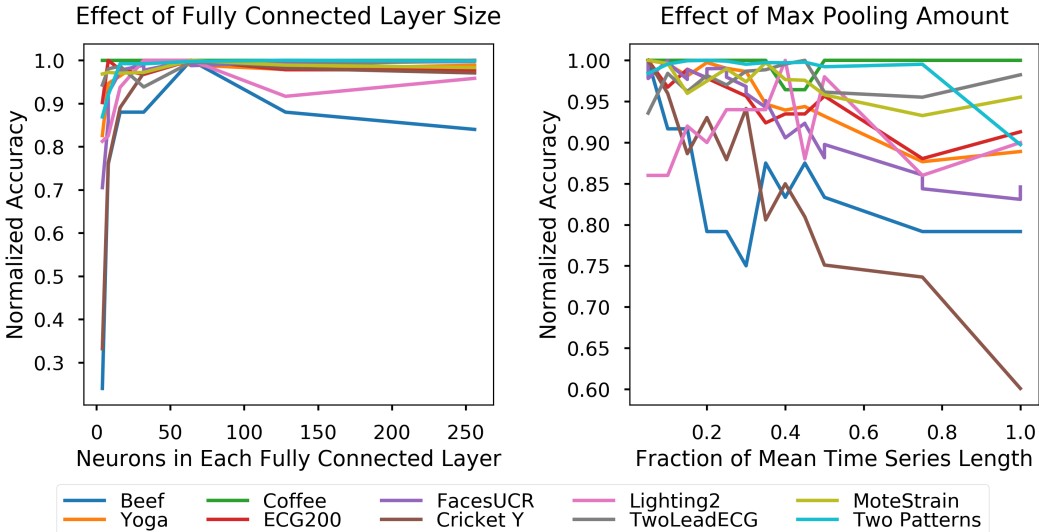

**Figure 2: Effect of fully connected layer size and degree of max pooling on model accuracy using held-out datasets. Even small fully connected layers and large amounts of max pooling— up to half of the length of the time series in some cases—have little or no effect on accuracy. For ease of visualization, each dataset's accuracies are scaled such that the largest value is 1.0.**

## 6.2 POOLING PERCENTAGE

The typical assumption in machine learning literature is that max pooling windows in convolutional architectures should be small to limit information loss. In contrast, time series algorithms often max pool globally across each example (e.g. (Grabocka et al., 2014)). Contrary to the implicit assumptions of both, we find that the level of pooling that results in the highest classification often falls in the 10-25% range, as shown by Figure 2.*right*

## 7 CONCLUSION

We present Jiffy, a simple and efficient metric learning approach to measuring multivariate time series similarity. We show that our method learns a metric that leads to consistent and accurate classification across a diverse range of multivariate time series. Jiffy's resilience to hyperparameter choices and consistent performance across domains provide strong evidence for its utility on a wide range of time series datasets.

Future work includes the extension of this approach to multi-label classification and unsupervised learning. There is also potential to further increase Jiffy's speed by replacing the fully connected layer with a structured (Bojarski et al., 2016) or binarized (Rastegari et al., 2016) matrix.

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
