# OpenReview forum: "Jiffy: A Convolutional Approach to Learning Time Series Similarity"
_ICLR.cc/2018/Conference — Reject_

### Official Review · AnonReviewer3 · 2017-11-24
**Well-written and conducted but limited technical contribution.**

**Rating:** 4
**Confidence:** 4

**Review:**

[Summary]

The paper is overall well written and the literature review fairly up to date.
The main issue is the lack of novelty.
The proposed method is just a straightforward dimensionality reduction based on
convolutional and max pooling layers.
Using CNNs to handle variable length time series is hardly novel.
In addition, as always with metric learning, why learning the metric if you can just learn the classifier?
If the metric is not used in some compelling application, I am not convinced.

[Detailed comments and suggestions]

* Since "assumptions" is the only subsection in Section 2,
I would use \texbf{Assumptions.} rather than \subsection{Assumptions}.

* Same remark for Section 4.1 "Complexity analysis".

* Some missing relevant citations:

Learning the Metric for Aligning Temporal Sequences.
Damien Garreau, Rémi Lajugie, Sylvain Arlot, Francis Bach.
In Proc. of NIPS 2014.

Deep Convolutional Neural Networks On Multichannel Time Series For Human Activity Recognition.
Jian Bo Yang, Minh Nhut Nguyen, Phyo Phyo San, Xiao Li Li, Shonali Krishnaswamy.
In Proc.  of IJCAI 2015.

Time Series Classification Using Multi-Channels Deep Convolutional Neural Networks
Yi ZhengQi LiuEnhong ChenYong GeJ. Leon Zhao.
In Proc. of International Conference on Web-Age Information Management.

Soft-DTW: a Differentiable Loss Function for Time-Series.
Marco Cuturi, Mathieu Blondel.
In Proc. of ICML 2017.

---

### Official Review · AnonReviewer2 · 2017-11-27
**A convolutional approach to learning time series similarities**

**Rating:** 8
**Confidence:** 3

**Review:**

Paper proposes to use a convolutional network with 3 layers (convolutional + maxpoolong + fully connected layers) to embed time series in a new space such that an Euclidian distance is effective to perform a classification. The algorithm is simple and experiments show that it is effective on a limited benchmark. It would be interesting to enlarge the dataset to be able to compare statistically the results with state-of-the-art algorithms. In addition, Authors compare themselves with time series metric learning and generalization of DTW algorithms. It would also be interesting to compare with other types of time series classification algorithms (Bagnall 2016) .

---

### Official Review · AnonReviewer1 · 2017-11-29
**Solid empirical analysis of a simple time series embedding technique**

**Rating:** 6
**Confidence:** 4

**Review:**

This paper presents a solid empirical analysis of a simple idea for learning embeddings of time series: training a convolutional network with a custom pooling layer that generates a fixed size representation to classify time series, then use the fixed size representation for other tasks. The primary innovation is a custom pooling operation that looks at a fraction of a sequence, rather than a fixed window. The experiments are fairly thorough (albeit with some sizable gaps) and show that the proposed approach outperforms DTW, as well as embeddings learned using Siamese networks. On the whole, I like the line of inquiry and the elegant simplicity of the proposed approach, but the paper has some flaws (and there are some gaps in both motivation and the experiments) that led me to assign a lower score. I encourage the authors to address these flaws as much as possible during the review period. If they succeed in doing so, I am willing to raise my score.

QUALITY

I appreciate this line of research in general, but there are some flaws in its motivation and in the design of the experiments. Below I list strengths (+) and weaknesses (-):

+ Time series representation learning is an important problem with a large number of real world applications. Existing solutions are often computationally expensive and complex and fail to generalize to new problems (particularly with irregular sampling, missing values, heterogeneous data types, etc.). The proposed approach is conceptually simple and easy to implement, faster to train than alternative metric learning approaches, and learns representations that admit fast comparisons, e.g., Euclidean distance.
+ The experiments are pretty thorough (albeit with some noteworthy gaps) -- they use multiple benchmark data sets and compare against strong baselines, both traditional (DTW) and deep learning (Siamese networks).
+ The proposed approach performs best on average!

- The custom pooling layer is the most interesting part and warrants additional discussion. In particular, the "naive" approach would be to use global pooling over the full sequence [4]. The authors should advance an argument to motivate %-length pooling and perhaps add a global pooling baseline to the experiments.
- Likewise, the authors need to fully justify the use of channel-wise (vs. multi-channel) convolutions and perhaps include a multi-channel convolution baseline.
- There is something incoherent about training a convolutional network to classify time series, then discarding the classification layer and using the internal representation as input to a 1NN classifier. While this yields an apples-to-apples comparison in the experiments, I am skeptical anyone would do this in practice. Why not simply use the classifier (I am dubious the 1NN would outperform it)? To address this, I recommend the authors do two things: (1) report the accuracy of the learned classifier; (2) discuss the dynamic above -- either admit to the reader that this is a contrived comparison OR provide a convincing argument that someone might use embeddings + KNN classifier instead of the learned classifier. If embeddings + KNN outperforms the learned classifier, that would surprise me, so that would warrant some discussion.
- On a related note, are the learned representations useful for tasks other than the original classification task? This would strengthen the value proposition of this approach. If, however, the learned representations are "overfit" to the classification task (I suspect they are), and if the learned classifier outperforms embeddings + 1NN, then what would I use these representations for?
- I am modestly surprised that this approach outperformed Siamese networks. The authors should report the Siamese architectures -- and how hyperparameters were tuned on all neural nets -- to help convince the reader that the comparison is fair.
- To that end, did the Siamese convolutional network use the same base architecture as the proposed classification network (some convolutions, custom pooling, etc.)? If not, then that experiment should be run to help determine the relative contributions of the custom pooling layer and the loss function.
- Same notes above re: triplet network -- the authors should report results in Table 2 and disclose architecture details.
- A stronger baseline would be a center loss [1] network (which often outperforms triplets).
- The authors might consider adding at least one standard unsupervised baseline, e.g., a sequence-to-sequence autoencoder [2,3].

CLARITY

The paper is clearly written for the most part, but there is room for improvement:

- The %-length pooling requires a more detailed explanation, particularly of its motivation. There appears to be a connection to other time series representations that downsample while preserving shape information -- the authors could explore this. Also, they should add a figure with a visual illustration of how it works (and maybe how it differs from global pooling), perhaps using a contrived example.
- How was the %-length pooling implemented? Most deep learning frameworks only provide pooling layers with fixed length windows, though I suspect it is probably straightforward to implement variable-width pooling layers in an imperative framework like PyTorch.
- Figure 1 is not well executed and probably unnecessary. The solid colored volumes do not convey useful information about the structure of the time series or the neural net layers, filters, etc. Apart from the custom pooling layer, the architecture is common and well understood by the community -- thus, the figure can probably be removed.
- The paper needs to fully describe neural net architectures and how hyperparameters were tuned.

ORIGINALITY

The paper scores low on originality. As the authors themselves point out, time series metric learning -- even using deep learning -- is an active area of research. The proposed approach is refreshing in its simplicity (rather than adding additional complexity on top of existing approaches), but it is straightforward -- and I suspect it has been used previously by others in practice, even if it has not been formally studied. Likewise, the proposed %-length pooling is uncommon, but it is not novel per se (dynamic pooling has been used in NLP [5]). Channel-wise convolutional networks have been used for time series classification previously [6].

SIGNIFICANCE

Although I identified several flaws in the paper's motivation and experimental setup, I think it has some very useful findings, at least for machine learning practitioners. Within NLP, there appears to be gradual shift toward using convolutional, instead of recurrent, architectures. I wonder if papers like this one will contribute toward a similar shift in time series analysis. Convolutional architectures are typically much easier and faster to train than RNNs, and the main motivation for RNNs is their ability to deal with variable length sequences. Convolutional architectures that can effectively deal with variable length sequences, as the proposed one appears to do, would be a welcome innovation.

REFERENCES

[1] Wen, et al. A Discriminative Feature Learning Approach for Deep Face Recognition. ECCV 2016.
[2] Fabius and van Amersfoort. Variational Recurrent Auto-Encoders. ICLR 2015 Workshop Track.
[3] Tikhonov and Yamshchikov. Music generation with variational recurrent autoencoder supported by history. arXiv.
[4] Hertel, Phan, and Mertins. Classifying Variable-Length Audio Files with All-Convolutional Networks and Masked Global Pooling.
[5] Kalchbrenner, Grefenstette, and Blunsom. A Convolutional Neural Network for Modelling Sentences. ACL 2014.
[6] Razavian and Sontag. Temporal Convolutional Neural Networks for Diagnosis from Lab Tests. arXiv.

---

> ### Comment · AnonReviewer1 · 2018-01-14
> **Good author response, increased score**
>
> I have increased my score by one point (from a 5 to a 6) to reflect the thoughtful and comprehensive response. The authors did a nice job of addressing many of my concerns about their methods and experiments, in particular:
>
> - they clarified that the the "constant number, variable width" pooling layer can be implemented in existing frameworks through appropriate padding of each time series. I accept that answer, though I want to encourage the authors to clarify what sort of padding they used (zero, repeat edge value, etc.), and I wonder about the impact of adding different amounts of padding to different length time series.
> - they showed in [10] that indeed, the KNN+embedding model beats the ConvNet used to generate the embeddings. With further thought, I've concluded that is not unexpected, as the KNN classifier on top of the embeddings provides a more flexible decision boundary than the linear classification output layer of the CovNet. That said, the small sample sizes of the data sets calls into question whether the differences are statistically significant.
> - they showed in [11] that Jiffy does beat max pooling, which is reasonable since the finer grained pooling necessarily preserves more detail
> - they showed in [11] that Jiffy also beats multichannel convolutions -- but I'd love some exposition hypothesizing why that might be
> - I think their explanation for why Jiffy beats Siamese/Triplet networks (small training data) is reasonable
> - I think the additional exposition re: motivation provided in their response would improve their paper if integrated. In particular, emphasizing retrieval (rather than classification) applications and providing experiments to back those up?
>
> I seriously considered a score of 7. However, I ultimately decided against that because a significant fraction of my critiques focused on the paper's exposition (motivation, related work, discussion). While the authors addressed some of this feedback in their response, they did not provide (as far as I can tell) a revised manuscript showing how they would integrate this new content into their actual paper. I understand that the ability to revise a submission is NOT the norm in the ML community and may be unfamiliar (or confusing) to first-time ICLR authors (it's also a bit of a chore over the holidays). Nonetheless, I feel uncomfortable (strongly) endorsing a submission that requires substantial rewrites.

---

> ### Comment · AnonReviewer1 · 2018-01-14
> **Comment on reproducibility report**
>
> I did in fact read the reproducibility report, but I want to clarify that I did NOT take it into account in my review (and for the record, I think its impact would have been negligible in this case). Reproduction during review represents an additional level of scrutiny for a submission, and while I feel that is Good Thing, I feel that it is unfair to take it into account during reviews if only SOME submissions are subject to that scrutiny.
>
> That said, I think two major points from the report are worth highlighting, one positive (+) and one negative (-):
>
> + by and large, the students were able to reproduce the results of the paper!
>
> - the students hypothesize (and I'm inclined to agree) that the results may be misleading due to the small size of the data, particularly the very small test sets. I will quote them:
>
> "We found that one of the most significant factors on accuracy was the random training and testing
> split, especially for the datasets that contained fewer examples...Our findings suggest that the accuracies obtained by Jiffy in the original paper are reasonable, but do not appear to be as robust to initialization and hyper-parameters as the
> authors claim."
>
> I recommend that the authors do something to quantify uncertainty of the performance measures, e.g., k-fold cross-validation or bootstrapping. I also encourage them to consider inclusion of at least one larger data set.
>
> The authors should also take note that the students assigned the original submission a "low to moderate [reproducibility] score." ;)

---

### Public Comment · (anonymous) · 2017-12-12
**Reproducing Results**


We are trying to replicate your results for the 2018 ICLR Reproducibility Challenge : http://www.cs.mcgill.ca/~jpineau/ICLR2018-ReproducibilityChallenge.html

1. Are you planning on releasing your code as described in the paper?  The repository is currently empty.

2.  Variable size maxpool implementation: Did you use padding? If so was it before or after the convolutional layer?

3. What initial learning rate was used for the Adam Optimizer?

4. Was a holdout evaluation set used?

---

> ### Author Response · Authors · 2017-12-12
> **Re: Reproducing Results**
>
> Thanks for your interest in our work! Here are the answers to your questions:
>
> 1. We will be releasing a fully testable implementation by the end of this weekend! Apologies for the delay; pre-processing and testing code will also be made available for ease of reproducibility.
> 2. We do use padding and this padding occurs prior to the convolutional layer. This code will be released as well.
> 3. The initial learning rate used is 2e-5.
> 4. Yes, a holdout evaluation set was used. For datasets with a pre-specified train/test split, we used those. For the other datasets, we partitioned 20% of the dataset for evaluation.
>
> Happy to answer additional questions about the implementation in the meantime.

---

> > ### Public Comment · (anonymous) · 2017-12-12
> > **Re: Reproducing Results**
> >
> > Thanks for the fast response!
> >
> > A few more questions:
> >
> > 1.What framework did you use to implement the net? - we have something running in tensorflow right now.
> >
> > 2. What were the stopping conditions were used for training?
> >
> > 3. Was a learning rate annealing schedule used?
> >
> > 4. Did you use non-default values for beta_1 and beta_2 for Adam?
> >
> > 5. Finally, what do you think explains the model outperforming siamese cnn and rnn?

---

> > > ### Author Response · Authors · 2017-12-12
> > > **Re: Reproducing Results**
> > >
> > > 1. We used tensorflow as well!
> > > 2. We train for a fixed number of iterations (20000).
> > > 3. We just used the built-in Adam optimizer, with the aforementioned learning rate.
> > > 4. No, we left beta_1 and beta_2 unmodified.
> > > 5. We believe the Siamese CNN/RNN perform poorly in comparison to our architecture due to its weakness in the face of multi-class problems and limited data. In n-class scenarios, the Siamese architecture attempts to specify O(n^2) constraints. This lines up with the Siamese CNN/RNN's poor performance on the Libras dataset - we hypothesize that this is a result of the combination of relatively little data (360 examples) and a high number of classes (15).
> > >
> > > We're hoping to get the code out earlier, before the deadline for this competition. Sunday is the absolute latest.

---

> > > > ### Public Comment · (anonymous) · 2017-12-13
> > > > **Re: Reproducing Results**
> > > >
> > > > Is there any chance you could put the preprocessing/cleaning code up earlier- we have something that is performing decently already- but it would be a major help to not have to load/clean each of the 16 datasets you used.

---

### Public Comment · ~Preston_Putzel1 · 2017-12-22
**Reproducibilty Report, and our code**

Here's the link to our git repo, which contains a reproducibility report, and our reimplementation code:

https://bitbucket.org/cagraff/iclr2018_repro_challenge_jiffy/src/cb8839bf6023?at=master

---

### Author Response · Authors · 2017-12-24
**Paper Clarifications & Further Experiments**

Thanks to all for the feedback - we believe the following experiments and discussion serve to strengthen the paper's argument and answer outstanding questions. Here we detail the paper's motivation and experimentation, compare to a number of suggested baselines, and elaborate on architecture-specific questions.

*Motivation and Experiment Choice*
The paper currently lacks clarity regarding our motivation/experiment choice in developing a distance metric with which to compare multivariate time series. This is partially due to our choice of experiment; we chose 1NN not because we expect it to be an especially good classifier, but because it is the standard means of evaluating time series representations and distance measures (Ding et. al. [1], Schafer et. al. [2], Shokoohi-Yekta et. al. [3]). Reviewer 1 and Reviewer 3 pose solid points regarding the value of considering an additional task. A distance metric is particularly valuable in the context of information retrieval. Constructing a compact representation for later retrieval of similar items is an important problem and has been the subject of a great deal of work, at least in the case of retrieving images. See [4], [5] for surveys, and [6],[7] for examples of doing so in the supervised setting.

For time series, consider the following use case. A hospital patient’s ECG signals indicate the presence of a premature ventricular contraction (PVC). To better understand the patient’s state, a physician would like to see similar heartbeats for comparison. Searching through a database of raw ECG waveforms would be computationally expensive, and, depending on the similarity measure, might not return useful examples. A learned distance measure could both accelerate the search through dimensionality reduction and increase the quality of the results. Note that, because heartbeat types are easily classified in most cases, the learning process can use labels. This clinical scenario occurs not only for ECG data, but for many medical time series [8]. We’ll articulate this use case and add another experiment to the paper.

The histogram linked at [12] visualizes the distribution of distances between same-label pairs and different-label pairs of examples. With this figure, we intend to show how DTW distances for different-class pairs are often the same as the DTW distances for same-class pairs. In the histogram, the blue bars describe the number of same-class example pairs that exist at that distance. The orange bars describe the same for different-class label pairs. Looking at AUSLAN and Libras, we see that the spread of distances for DTW is larger in both cases, where blue bars and orange bars often exist for the same distance. Comparing the DTW histograms to the Jiffy histograms, we see that the Jiffy histogram of distances has a smaller spread of distances for same-class pairs while achieving higher nearest-neighbor classification accuracy.

*Baselines*
We support our method with the comparison of the NN classifier to three baselines: the basic CNN, a global pooling baseline, and a multi-channel CNN baseline.

Linked at [10] is a figure demonstrating the comparison of the NN classifier to the CNN's softmax classification accuracies. For 5/6 of the datasets, the 1NN classification accuracy exceeds the CNN classifier's accuracy. As expected, the addition of more neighbors in the 3NN and 5NN results serves to increase or maintain the accuracy of each dataset, save for ECG and Libras. This may be explained by how small the ECG dataset is - points at the border of each cluster are often close enough to a separate cluster to adopt its points.

Linked at [11] is a figure comparing the performance of the global pooling baseline and multi_channel CNN baseline to Jiffy. Both the global pooling and multi-channel convolution baselines fail to perform as consistently as the Jiffy’s percentage-pooled, single-channel convolution architecture. The multi-channel convolution baseline often achieves comparable accuracy, but this fails to justify the extra parameters in this model.

*Architecture*
The architecture used in the Siamese/triplet network comparisons is identical to the architecture of the proposed CNN classifier. The Siamese RNN architecture is based on the architecture proposed by Pei et. al. [9]. We will elaborate on how the hyper-parameters were specified for each of these networks.

The architecture is parameterized by the %-pooling necessary. Because we know the (padded) length of the time series at the start of training, however, we can create pooling layers of “constant” size using existing APIs. E.g., for length 100 time series and 15% pooling, we simply use a pooling size of 15.

With respect to Figure 1, we intend to clarify the %-pooling aspect of the architecture - no hyperparameters were tuned for the resulting architecture, a detail that we will communicate in the revised edition of the paper.

---

> ### Author Response · Authors · 2017-12-24
> **Citations**
>
> [1] Ding, Hui, et al. "Querying and mining of time series data: experimental comparison of representations and distance measures." Proceedings of the VLDB Endowment 1.2 (2008): 1542-1552.
>
> [2] Schäfer, Patrick. "Towards time series classification without human preprocessing." International Workshop on Machine Learning and Data Mining in Pattern Recognition. Springer, Cham, 2014.
> APA
>
> [3] Shokoohi-Yekta, Mohammad, Jun Wang, and Eamonn Keogh. "On the non-trivial generalization of dynamic time warping to the multi-dimensional case." Proceedings of the 2015 SIAM International Conference on Data Mining. Society for Industrial and Applied Mathematics, 2015.
>
> [4] Wang, Jingdong, et al. "A survey on learning to hash." IEEE Transactions on Pattern Analysis and Machine Intelligence (2017).
>
> [5] Wang, Jun, et al. "Learning to hash for indexing big data—a survey." Proceedings of the IEEE 104.1 (2016): 34-57.
>
> [6] Erin Liong, Venice, et al. "Deep hashing for compact binary codes learning." Proceedings of the IEEE Conference on Computer Vision and Pattern Recognition. 2015.
>
> [7] Zhang, Ruimao, et al. "Bit-scalable deep hashing with regularized similarity learning for image retrieval and person re-identification." IEEE Transactions on Image Processing 24.12 (2015): 4766-4779.
>
> [8] Kim, Yongwook Bryce, and Una-May O'Reilly. "Large-scale physiological waveform retrieval via locality-sensitive hashing." Engineering in Medicine and Biology Society (EMBC), 2015 37th Annual International Conference of the IEEE. IEEE, 2015.
>
> [9] Pei, Wenjie, David MJ Tax, and Laurens van der Maaten. "Modeling time series similarity with siamese recurrent networks." arXiv preprint arXiv:1603.04713 (2016).
>
> [10] https://ibb.co/g2X3LR
>
> [11] https://ibb.co/kTRkZm
>
> [12] https://ibb.co/iijkN6

---

### Decision · Program_Chairs · 2018-01-29
**ICLR 2018 Conference Acceptance Decision**

**Decision:**

Reject

**Comment:**

R1 was neutral on the paper: they liked the problem, simplicity of the approach, and thought the custom pooling layer was novel, but raised issues with the motivation and design of experiments. R1 makes a reasonable point that training a CNN to classify time series, then throw away the output layer and use the internal representation in 1-NN classification is hard to justify in practice.
Results of the reproducibility report were good, though pointed out some issues around robustness to initialization and hyper-parameters. R2 gave a very strong score, though the review didn’t really expound on the paper’s merits. R3 thought the paper was well written but also sided with R1 on novelty. Overall, I side with R1 and R3. Particularly with respect to the practicality of the approach (as pointed out by both these reviewers). I would feel differently if the metric was used in another application beyond classification.